# Performance Improvement of PVDF–HFP-Based Gel Polymer Electrolyte with the Dopant of Octavinyl-Polyhedral Oligomeric Silsesquioxane

**DOI:** 10.3390/ma14112701

**Published:** 2021-05-21

**Authors:** Xin Guo, Shunchang Li, Fuhua Chen, Ying Chu, Xueying Wang, Weihua Wan, Lili Zhao, Yongping Zhu

**Affiliations:** 1School of Chemical Engineering, University of Chinese Academy of Sciences, Beijing 100049, China; guoxin181@mails.ucas.ac.cn (X.G.); lishunchang19@mails.ucas.ac.cn (S.L.); chenfuhua19@mails.ucas.ac.cn (F.C.); 2State Key Laboratory of Multiphase Complex Systems, Institute of Process Engineering, Chinese Academy of Sciences, Beijing 100190, China; ychu@ipe.ac.cn (Y.C.); xywang@ipe.ac.cn (X.W.); 3State Key Laboratory of Advanced Chemical Power Sources (SKL-ACPS), Guizhou 563003, China; wkhujing@wfust.edu.cn

**Keywords:** lithium-ion batteries, gel polymer electrolytes, ionic conductivity, OVAPOSS, PVDF–HFP

## Abstract

Gel polymer electrolytes have the advantages of both a solid electrolyte and a liquid electrolyte. As a transitional product before which a solid electrolyte can be comprehensively used, gel polymer electrolytes are of great research value. They can reduce the risk of spontaneous combustion and explosion caused by leakage during the use of conventional liquid electrolytes. Poly(vinylidene-fluoride-co-hexafluoropropylene) (PVDF–HFP), a material with excellent performance, has been widely utilized in the preparation of gel polymer electrolytes. Here, PVDF–HFP-based gel polymer membranes with polyvinyl pyrrolidone (PVP) pores were prepared using a phase inversion method, and Octavinyl-polyhedral oligomeric silsesquioxane (OVAPOSS) was doped to improve its temperature resistance as well as its ionic conductivity, to enhance its safety and electrochemical performance. The final prepared polymer membrane had a porosity of 85.06% and still had a certain mechanical strength at 160 °C without any shrinkage. The gel polymer electrolyte prepared with this polymer membrane had an ionic conductivity of 1.62 × 10^−3^ S·cm^−1^ at 30 °C, as well as an electrochemical window of about 5.5 V. The LiCoO_2_-Li button half-cell prepared therefrom had a specific capacity of 141 mAh·g^−1^ at a rate of 1C. The coulombic efficiency remained above 99% within 100 cycles and the capacity retention rate reached 99.5%, which reveals an excellent cycling stability.

## 1. Introduction

Since the invention of lithium-ion batteries in the 1870s, there has been significant development in various electronic mobile devices, due to their high specific capacity and portability which, in turn, have promoted the further development of the lithium-ion battery industry. Accompanying the large-scale use of high-capacity electrical appliances, lithium-ion batteries have been applied in more scenarios, and more demands have been made for their useability, such as a higher voltage, higher capacity, longer cycle life, higher safety, etc. [1,2], which puts forward more stringent requirements on the materials of lithium-ion batteries. The existing lithium-ion battery system mainly uses liquid electrolytes. Polyolefin is used as the membrane, which may suffer from heat shrinkage at high temperatures and leakage of the liquid electrolyte, leading to many safety hazards during the use of the batteries [3,4].

Gel polymer electrolytes (GPEs) combine the characteristics of solid electrolytes and liquid electrolytes. Due to their good ionic conductivity (room temperature >10^−4^ S·cm^−1^), wide electrochemical window, and good electrode compatibility, GPEs have attracted much attention [5,6,7].Poly(1,1-difluoroethylene) (PVDF), an excellent material, has the advantages of corrosion resistance, temperature resistance, easy film formation, and a high dielectric constant. Due to its –C–F_2_-group with a strong electronegativity, PVDF is able to greatly promote the dissociation of lithium salts in the electrolyte. Moreover, the –C–F-bond has a high resistance to electrochemical oxidation and superior electrochemical stability, so the GPEs prepared by it demonstrate excellent performance [8,9]. However, PVDF is a crystalline compound, and when the membrane prepared by it has a relatively high crystallinity, the electrolyte that can be stored during activation will decrease. Therefore, PVDF–HFP (PVDF with hexafluoropropylene copolymerization) has been developed [10]. The crystallinity of PVDF–HFP is much lower than that of PVDF alone and the GPEs prepared by PVDF–HFP display a higher room temperature conductivity, so it is being widely applied in lithium-ion batteries [11].

GPEs have relatively poor mechanical properties and thermal stability. Generally, they are modified by doping with inorganic fillers [12]. Inorganic fillers (such as SiO_2_ [13,14,15,16], TiO_2_ [17], Al_2_O_3_ [6,18], ZnO [19], etc.) doped into the polymer membrane are able to effectively improve all aspects of the membrane performance. The oxygen atom system on the oxide filler can act as a Lewis base and interact with Lewis acid Li^+^ to form a new ionic transferring channel, which enables the membrane to have a higher room temperature ionic conductivity and lithium transport number. Also, its high dielectric constant can promote the dissociation of lithium salt, which increases the number of free carriers in the system. Moreover, when these nanoparticles can act as nodes of the internal network structure of the membrane, they can effectively disperse the stress of the membrane and improve the mechanical properties, as well as the thermal stability of the polymer. However, agglomeration will occur during the use of inorganic fillers, so polyhedral oligomeric silsesquioxane (POSS), an organic–inorganic composite material, can be considered as an additive.

POSS owns an inorganic silica core and eight evenly distributed positions on the corners of the polyhedron, which serves to combine with organic functional groups to meet different application scenarios. Seen as a material with excellent performance, POSS has been widely utilized and studied in recent years. Its enhancement and flame-retardant effects on polymers have a great influence on the safety of lithium-ion batteries. Shang, D. et al. [20] made use of POSS materials modified with ionic liquid to make a novel solid electrolyte that could be used at high temperatures. Through electrochemical performance testing and cyclic performance testing, it was found that the ionic conductivity was relatively high, greater than 10^−4^ S·cm^−1^, which had a good cyclic stability as well as great potential for development. Octavinyl-polyhedral oligomeric silsesquioxane (OVAPOSS) can be dissolved in a small amount of organic reagents (*N*,*N*–Dimethylformamide (DMF)) and be introduced into the PVDF–HFP membrane by blending to improve the mechanical strength, temperature resistance, and room temperature ionic conductivity of the polymer. The eight vinyl groups of OVAPOSS enable it to be evenly inlaid between the polymer segments and to have a certain binding capacity.

Here, OVAPOSS and PVDF–HFP were evenly mixed by a simple phase inversion method so OVAPOSS would be able to play an enhanced role around the organic segment of PVDF–HFP. Moreover, the additive Polyvinylpyrrolidone (PVP) was introduced to adjust the pore structure of the membrane to improve the liquid absorption ability of the polymer membrane and to enhance the electrochemical performance of it.

## 2. Materials and Methods

### 2.1. Materials

Polyvinylidene fluoride-hexafluoropropylene (PVDF–HFP) was purchased from Arkema, France. Octavinyl-polyhedral oligomeric silsesquioxane (OVAPOSS), polyvinylpyrrolidone (PVP–K30), *N*–Methyl–2–Pyrrolidone (NMP), *N*,*N*–Dimethylformamide (DMF), *N*–butanol (C_4_H_10_O) were all analytically pure, bought from Aladdin (Shanghai) Reagents Co., Ltd, Shanghai, China. Lithium tablets (Li), LiCoO_2,_ and electrolytes (1 M LiPF_6_ (ethylene carbonate (EC): dimethyl carbonate (DMC): ethyl methyl carbonate (EMC) = 1:1:1, *v*/*v*)) were bought from Guangdong Candle New Energy Technology Co., Ltd, Guangdong, China. Acetylene black, binder polyvinylidene fluoride (PVDF), and aluminum foil were purchased from Hefei Crystalgen Material Technology Co., Ltd, Hefei, China.

### 2.2. Preparation of PVDF–HFP Polymer Membrane with OVAPOSS

PVDF–HFP was baked in an oven at 80 °C for 12 h, then 5 g were removed and put into a beaker and 500 mg of PVP were added; took a certain quantity of OVAPOSS (with a mass ratio of 2, 5, and 7% to PVDF–HFP) was taken and fully dissolved by 25 g DMF, then poured into the aforementioned beaker containing PVDF–HFP. The beaker was sealed and heated in a water bath at 50 ℃ then the contents stirred for 4 h to form a uniform membrane-forming solution. Next, the inner air of the membrane liquid was removed by ultrasonic vacuum, etc. to obtain a colorless, clear, and transparent homogeneous solution. Using a membrane maker with a 200-micron slit, the membrane liquid was evenly coated on a flat and smooth glass plate to form a wet film. After coating, the film was left standing in the air for 30 s. Subsequentially, the glass plate coated with the wet film was put into deionized water for phase inversion, then removed after 24 h. Next, the moisture on the PVDF–HFP surface was wiped off with filter papers, the film dried at room temperature, and cut into round pieces. It was then dried in an oven at 80 °C for 24 h and placed in a glove box for later use.

The dried membrane was then soaked in electrolytes for 2 h to activate the gel and, then, we assembled the membrane into a button cell for electrochemical performance tests.

Through changing the additional amounts of OVAPOSS, its impact on the performance of the membrane could be explored. Four groups of systems were set with no OVAPOSS added, 2% OVAPOSS added, 5% OVAPOSS added, and 7% OVAPOSS added, respectively. The prepared membranes were named as follows: P_1_D_5_P_10%_, P_1_D_5_O_2%_P_10%_, P_1_D_5_O_5%_P_10%,_ and P_1_D_5_O_7%_P_10%_.

### 2.3. Preparation of Positive Plate and Assembly of Button Cell

We took 0.85 g of lithium cobalt oxide-positive materials (nominal specific capacity of 140 mAh/g), then mixed and ground it with 0.1 g of acetylene black for 20–30 min. Then, we added 1.4916 g of the PVDF binder solution with a mass percentage of 5% (the solvent was NMP) and rapidly ground it (if the slurry became dry, NMP could be added proportionately). We applied the slurry to an aluminum foil, dried it in a blast drying oven, and cut it into 12 mm round pieces for later use. The active substance content of the positive plate required weighing before assembling the battery. The battery assembly needed to be conducted in a glove box with the content of water and oxygen below 0.1 ppm.

The button cell was assembled in the order of negative shell, lithium tablet, gel polymer electrolyte, positive plate, gasket, shrapnel, and positive shell. The installed cell needed to stand for some time before the subsequent tests could be performed.

### 2.4. Characterization of Polymer Membrane

A scanning electron microscope (SEM) was used to observe the microstructure of the polymer membrane (JSM-7800 (Prime) cold-field emission scanning electron microscope from JEOL Co., Ltd, Tokyo, Japan. The membrane is a non-conductive material. The cross-section was obtained using a liquid nitrogen brittle fracture treatment and the platinum was sprayed before observation. The thickness of the sample was measured by an electronic digital outside micrometer. Physical analysis of the synthesized materials was carried out via an X-ray in situ diffractometer (XRD) from Japan Science Co., Ltd, Kyoto, Japan. A Shimadzu tensile testing machine, AGS-X, Shimadzu Co., Ltd, Kyoto, Japan, was used to conduct the mechanical performance tests of the polymer membrane to test the stress–strain curve of it. The samples were cut into strips, with a width of about 5 mm and a length of about 50 mm, and we set a tensile rate of 20 mm/min. The different groups of polymer membranes were cut into 2 × 2 cm^2^ pieces, let stand for 2 h for each rise of 10 °C in the temperature range from 90 to 160 °C, and we observed the morphological changes at the different temperatures to investigate their thermal stability. The porosity of the polymer membrane can be calculated by the following Equation (1):(1)$$k\left(\%\right)=\frac{W_{2}-W_{1}}{\rho_{L}V}\times 100\%$$
where, ρ_L_ is the density of *N*–butanol, W_1_ and W_2_ are the mass before and after soaking in *N*-butanol for 6 h, respectively. The polymer membrane was cut into rectangular pieces, whose thickness and area were measured to calculate the volume V. The liquid absorption rate (η) of the polymer membrane can be calculated by the following Equation (2):(2)$$\eta \left(\%\right)=\frac{W_{t}-W_{o}}{W_{o}}\times 100\%$$
where W_o_ and W_t_ stand for the mass of the polymer membrane before and after soaking in liquid electrolytes for 6 h, respectively.

### 2.5. Electrochemical Performance Test

The specific test method of ionic conductivity was as follows: the polymer membranes were cut into 16 mm round pieces, soaked, and activated in the electrolyte for 2 h until they became transparent gel-like polymer electrolytes. Then, the electrolytes were placed between two stainless steel spacers and assembled into an SS (Steel Sheet)/gel polymer electrolyte/SS blocking battery. After standing for 4 h, we scanned the blocked battery with an electrochemical workstation in the range of 0.1–10^5^ Hz. Then we calculated the ionic conductivity of the polymer membrane using Equation (3).
(3)$$\sigma =\frac{d}{R_{b}\times S}$$
where R_b_ stands for the bulk resistance blocking the battery; S for the effective area of the polymer membrane; and d for the thickness of the membrane.

The ionic conductivity of the GPEs at each temperature were obtained by measuring the bulk resistance of the blocked battery at 30, 40, 50, 60, 70, and 80 °C, respectively. We tested the relationship between the ionic conductivity and temperature, and then used the Arrhenius Equation (4), to conduct the fitting calculation of the activation energy of the GPEs.
(4)$$\sigma =\sigma_{0}\exp\left(-\frac{E_{a}}{\mathrm{RT}}\right)$$
where σ is the ionic conductivity, R is the gas constant, T is the test temperature, and E_a_ is the activation energy.

Using lgσ as the ordinate and 1/T as the abscissa, a straight line was obtained with the slope being −E_a_/R, the help from which the activation energy of the GPEs could be calculated. Generally speaking, the lower the E_a_ was, the better the ion transmission ability was [21,22].

The electrochemical window test was to cut the polymer membrane into 16 mm round pieces, soak them in 1 M LiPF6 electrolytes, then place it between the stainless-steel gasket and the lithium sheet, and assemble it into SS/gel polymer electrolyte/Li semi-blocking batteries. After standing for 4 h, we scanned it with a sweep speed of 0.5 mV/s at the open circuit voltage of ~6 V to obtain the Linear Sweep Voltammetry (LSV) diagram of the voltage and current.

The Li_2_CoO_2_/gel polymer electrolyte/Li half-cell assembled by GPEs was used to conduct the constant current charge and discharge test by using the Wuhan Land CT2001A test system. The battery was activated three times at a current density of 0.1 C (1 C = 140 mAh/g) in a constant current mode, and then charged and discharged at a current density of 1 C in the voltage range of 2.8–4.3 V. Finally, we characterized the cycle stability and charge-discharge efficiency of the polymer membrane materials.

## 3. Results and Discussions

### 3.1. Physical Characterization of Polymer Membrane

Figure 1 shows the X-ray diffraction (XRD) patterns of the membranes prepared with different contents of OVAPOSS. The XRD pattern of the P_1_D_5_P_10%_ membrane without OVAPOSS added displayed two distinct diffraction peaks when 2θ = 18.51 and 20.09°, indicating that the membrane mainly contained α phase. After adding OVAPOSS, its peak intensity at 18.51 and 20.09° decreased, showing that the addition of OVAPOSS could reduce the crystallinity of the membrane. Comparing the XRD patterns of the pure OVAPOSS reagent and the OVAPOSS membrane, it can be found that both the P_1_D_5_O_5%_P_10%_ membrane and the P_1_D_5_O_7%_P_10%_ membrane had obvious OVAPOSS peaks when 2θ = 9.8, 23.2, and 27.5° [23], which showed that OVAPOSS had been successfully doped into the membrane.

Figure 2 shows the surface and cross-sectional SEM images of polymer membranes prepared with different amount of OVAPOSS additions. When OVAPOSS was added to 2%, the pores on the upper and lower surfaces of the membrane were evenly distributed. The upper surface had larger pores, while the lower surface appeared with a fish-scale structure and the internal pores were abundant. Compared to the P_1_D_5_P_10%_ membrane, the addition of OVAPOSS was able to promote the development of pores on the upper and lower surfaces. This was because the addition of OVAPOSS reduced the viscosity of the system and further reduced the surface strength of the membrane during the phase inversion. The phase inversion was completed sufficiently, so the PVP in the system could be completely converted out. When the content of OVAPOSS increased to 5%, the pores on the upper surface of the membrane disappeared and many gullies showed up. Meanwhile, there were many small particles of OVAPOSS evenly distributed on the surface, and the pores on the lower surface also were evenly distributed, with a pore size of about 0.1 microns. The membrane was rich in pores inside, with a lot of large pores whose diameters were about 20 microns, which provided much space for the storage of electrolytes. The appearance of large pores indicated that more PVP in the system had been transferred out, which was due to the fact that the unique organic–inorganic composite structure of OVAPOSS played an excellent role in its distribution in the organic membrane. The distribution of OVAPOSS on the surface provided mechanical support for the upper surface layer so the phase inversion could be conducted more adequately, and the membrane would not collapse due to the excessive pore size. When the additional amount of OVAPOSS continued to increase to 7%, the pore size of the lower surface of the membrane increased, and the large pores inside the membrane were more fully developed. Comparing Figure 2d,e, it can be found that the membrane with 5% addition had more large pores than the membrane with 7% addition, and the distribution also was more even.

Figure 3 consists of histograms of porosity and the liquid absorption rates of polymer membranes prepared with different OVAPOSS additions. It shows, with the increase in OVAPOSS, the porosity and liquid absorption rate of the membrane increased first and then decreased. When the additional amount was 5%, the porosity reached the maximum of 85.06% and the liquid absorption rate reached 345.36%.

Compared with the P_1_D_5_P_10%_ membranes, the porosity and liquid absorption rate of both the P_1_D_5_O_5%_P_10%_ and P_1_D_5_O_7%_P_10%_ membranes were greatly improved. The addition of OVAPOSS promoted the increase in the porosity of the membrane. This was because the addition of OVAPOSS decreased the viscosity of the casting liquid, and more PVP was transferred out to increase the number of large pores inside the membrane. However, when the amount of OVAPOSS reached 7%, both the porosity and liquid absorption rate decreased, which resulted from the distribution of OVAPOSS on the surface layer had gradually migrated to the interior of the membrane, reducing the porosity inside it. The high porosity of the polymer membrane and its combined structure of large and small pores enabled it to have good electrolyte adsorption and storage capabilities, which could improve its ionic conductivity at room temperature. When used in lithium-ion batteries, it also could improve the battery cycle performance and extend the battery life.

Figure 4 shows the mechanical strength of polymer membranes prepared with different amounts of OVAPOSS. It was found, when the addition of OVAPOSS increased from 0 to 2%, the tensile strength decreased from 26.37 to 3.7 MPa, and the elongation at the break increased from 39.05 to 135.42%. Comparing the P_1_D_5_ membrane with the P_1_D_5_O_2%_P_10%_ membrane, both the tensile strength and elongation at break of the two were similar. Considering the aforementioned morphology and porosity analyses, there were many pores increasing inside the P_1_D_5_O_2%_P_10%_ membrane. This was because the addition of a small amount of OVAPOSS reduced the viscosity of the casting liquid system. During the process of phase inversion, more PVP inside the wet membrane was dissolved and removed by water, thereby forming more pores inside the membrane. Therefore, the mechanical strength of the P_1_D_5_O_2%_P_10%_ membrane was lower than that of the P_1_D_5_ membrane.

Accompanying the increase in OVAPOSS, the tensile strength of the membrane with a 5% addition increased to 7.23 MPa, and the elongation at break decreased to 83.09%, which was stronger than that of the P_1_D_5_ membrane. Due to the uniform distribution of OVAPOSS on the upper and lower surfaces of the membrane, the internal network structure of the membrane was supported. OVAPOSS particles were dispersed inside the membrane and formed the nodes of the 3D network in the PVDF–HFP network, which improved the ability of the membrane to disperse stress; therefore, the mechanical properties would be enhanced. When adding OVAPOSS to 7%, the tensile strength of the membrane dropped to 2.74 MPa, and the elongation at break dropped to 70.94%. Excessive addition of OVAPOSS agglomerated inside the system, destroying the network structure of the system. Then the uniformity of the membrane became worse, as did the mechanical properties. Thus, when OVAPOSS was added to 5%, it exhibited the best mechanical modification performance.

Figure 5 is the thermogravimetric (TG) curve of each membrane in an argon atmosphere. It was found that the membrane was relatively stable before 430 °C. After 430 °C, with the continuous increase of the temperature, the membrane would be thermally decomposed. At 550 °C, the membrane was basically completely decomposed. Figure 6 is a comparison diagram of the thermal shrinkage of each group of membranes. Each group of membranes could maintain the original shape at 120 °C. When the temperature was raised to 130 °C, the polypropylene (PP) membrane shrank to 60% of its original size, and it could no longer play the role as a membrane, while there was no significant change in the membranes of the other groups. When the temperature rose to 140 °C, the PP membrane further shrank, and the color changed from white to transparent. The P_1_D_5_P_10%_ membrane also changed from translucent to transparent. However, there was basically no obvious change in the other groups. When the temperature rose to 150 °C, the P_1_D_5_ membrane and the P_1_D_5_P_10%_ membrane ruptured, while the sizes of the P_1_D_5_O_5%_P_10%_ membrane and the P_1_D_5_O_7%_P_10%_ membrane basically did not change, and the color of them turned slightly yellow. This was because the OVAPOSS filler in the membrane generated SiO_2_ in the oxygen environment, and a protective layer was formed on the surface of the polymer to prevent the shrinkage of the polymer membrane, thereby ensuring the safety of the battery at high temperatures. The normal operating temperature of lithium-ion batteries is below 80 °C, indicating that the PVDF–HFP-based gel electrolyte can be used safely.

### 3.2. Electrochemical Characterization of Polymer Membranes

Figure 7 is the room-temperature impedance diagram of the polymer membrane. The bulk resistor of each membrane could be obtained from the intercept, and then the room-temperature ionic conductivity of each membrane could be calculated according to Equation (3). Table 1 shows the thickness, room-temperature resistance, as well as the conductivity of each polymer membrane. The P_1_D_5_P_10%_ polymer membrane with PVP added was thicker than the P_1_D_5_ membrane and also had a higher ionic conductivity, which was caused by the residue of PVP in the membrane. The O in the C=O bond of PVP could act as a Lewis base in the membrane and paired with Lewis acid Li^+^ to promote the dissociation of the lithium salt. The polymer membrane doped with 2% OVAPOSS on the basis of the P_1_D_5_P_10%_ polymer membrane was very thin. The membrane would be squeezed when assembled into a battery. Its excessive porosity and low mechanical strength made it unable to withstand squeezing and it would then deform, resulting in a low liquid holding capacity in actual use. Therefore, its ionic conductivity was relatively lower than that of the P_1_D_5_P_10%_ membrane. When the doping amount of OVAPOSS increased to 5%, the mechanical strength of the polymer membrane would increase, and it could withstand a certain pressure. Also, the liquid holding capacity of it would increase and the room temperature ionic conductivity of the P_1_D_5_O_5%_P_10%_ membrane would increase to 1.62 × 10^−3^ S·cm^−1^. While the doping amount of OVAPOSS continuously increased, the room temperature ionic conductivity of the membrane decreased again, which was 1.16 × 10^−3^ S·cm^−1^. This showed that the addition of OVAPOSS was able to improve the ionic conductivity of the membrane. However, the excessive addition of OVAPOSS would be easy to agglomerate in the polymer matrix. The large-volume aggregated OVAPOSS molecules impeded the transfer of lithium ions in the system, thereby reducing the conductivity of the polymer electrolyte.

After analyzing the ionic conductivity of different membranes at room temperature, the ionic conductivity of membranes doped with different contents of OVAPOSS in the temperature range of 20–80 °C was compared. Figure 8a is the ionic conductivity diagram of the membranes prepared with different doping amounts of OVAPOSS at different temperatures. According to Figure 8a, the ionic conductivity of the membrane gradually increased as the temperature rose, which was caused by the increase in the movement velocity of the lithium ions due to the increase in temperature. The ionic conductivity data of each group of membranes at different temperatures are shown in Table 2. Figure 8b is the Arrhenius conductivity map of each group of polymer membranes, and the ionic conduction behavior conformed to the Arrhenius equation within the test temperature range. According to Equation (4), the activation energy of the membranes prepared with various amounts of OVAPOSS could be calculated, and the specific data are shown in Table 3. The activation energy of the polymer membrane with OVAPOSS was significantly reduced, indicating that this kind of filler had a promoting effect on the transportation of lithium ions. However, the activation energy increased with the addition of 7% OVAPOSS, showing that the excessive content of OVAPOSS would agglomerate and block the transmission channel of lithium ion, which was not conducive to the transport of lithium ions.

The electrochemical stability window is an important parameter to measure the electrochemical stability of the membrane. Figure 9 is a linear sweep volt-ampere curve of the membrane at room temperature. It can be found from the Figure that the electrochemical window of the commercial membrane was around 4.2 V, and that of the membrane with 5% POSS was around 5.5 V. Therefore, the membrane with 5% POSS had better electrochemical stability than the commercial membrane. 

Figure 10 is the cyclic voltammetry curve of a LiCoO_2_-Li half-cell assembled by a P_1_D_5_O_5%_P_10%_ gel polymer electrolyte. It can be seen from the Figure that the basic redox peaks of the four-cycle voltammetry curves of the battery coincide, indicating that the battery had a good cycle stability.

Figure 11a shows the cyclic performance of the button cell prepared by polymer membranes with a current density of 1 C at room temperature. According to the Figure, the button cell assembled with OVAPOSS-added membrane still had a capacity retention rate of 99% after 100 cycles. Therefore, a membrane with OVAPOSS added possessed a greater advantage in maintaining the battery capacity and the stability of charging and discharging.

Figure 11b is the first charge and discharge diagram of button cells prepared by each group of polymer membranes at room temperature. It can be drawn from the Figure that the first discharge-specific capacity of the battery using a PP membrane was 126 mAh·g^−1^; that of the battery using a P_1_D_5_O_7%_P_10%_ membrane was 134 mAh·g^−1^, and that of the battery using a P_1_D_5_O_5%_P_10%_ membrane was 141 mAh·g^−1^. Therefore, the specific capacity of batteries assembled with the OVAPOSS membrane in the same range of charge and discharge voltage was higher than that of cells assembled with just the PP membrane. This was because the addition of OVAPOSS effectively increased the ionic conductivity of the membrane, which reduced the energy consumption of the lithium-ion transmission inside the battery and reduced the energy loss. When the additional amount of OVAPOSS reached 5%, the battery was able to possess the most stable charge-discharge cycle performance, the highest specific capacity, as well as the best performance.

## 4. Conclusions

Here, PVDF–HFP, PVP, and OVAPOSS were used as raw materials to successfully prepare porous polymer membranes with a high performance and high safety using a phase inversion method. XRD shows that OVAPOSS can be successfully doped into PVDF–HFP membranes, which enables them to effectively reduce the crystallinity of the system. Adding OVAPOSS can reduce the viscosity of the system and improve the porosity and liquid absorption rate of the prepared membrane. When the doping amount of OVAPOSS reached 5%, the polymer membrane showed the best performance with the maximum porosity of 85.06%, liquid absorption rate of 345.36%, mechanical strength of 7.23 MPa, elongation at break of 83.09%, and ionic conductivity of 1.62 × 10^−3^ S·cm^−1^ at room temperature. The electrochemical window measured by the LSV curve was greater than 5.5 V. The doping of OVAPOSS significantly improves the thermal stability of the PVDF–HFP membrane, preventing shrinkage at 160 °C, maintaining a certain strength, and improving the thermal safety of the battery. The Li_2_CoO_2_/gel polymer electrolyte/Li half-cell prepared using P_1_D_5_O_5%_P_10%_ GPE had a first discharge capacity of 141 mAh·g^−1^ at a current density of 1 C, and the capacity retention rate after 100 cycles was 99.5%, which displays an excellent electrochemical performance and cycle stability. 

## Figures and Tables

**Figure 1 materials-14-02701-f001:**
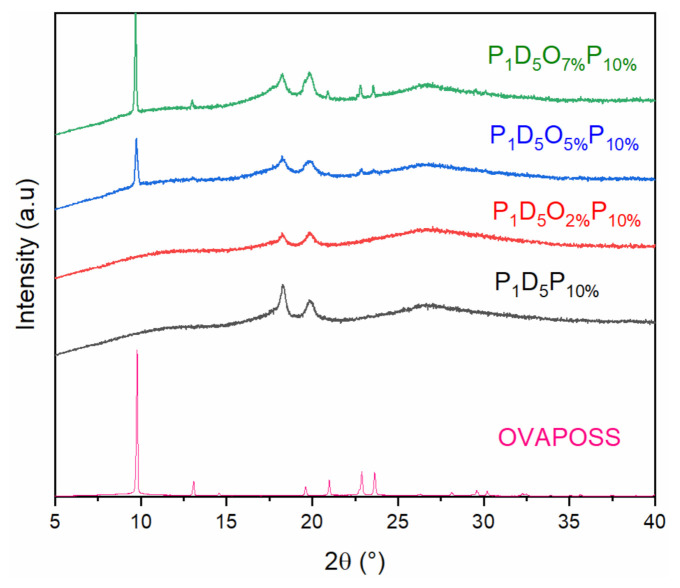
XRD curve of polymer membranes.

**Figure 2 materials-14-02701-f002:**
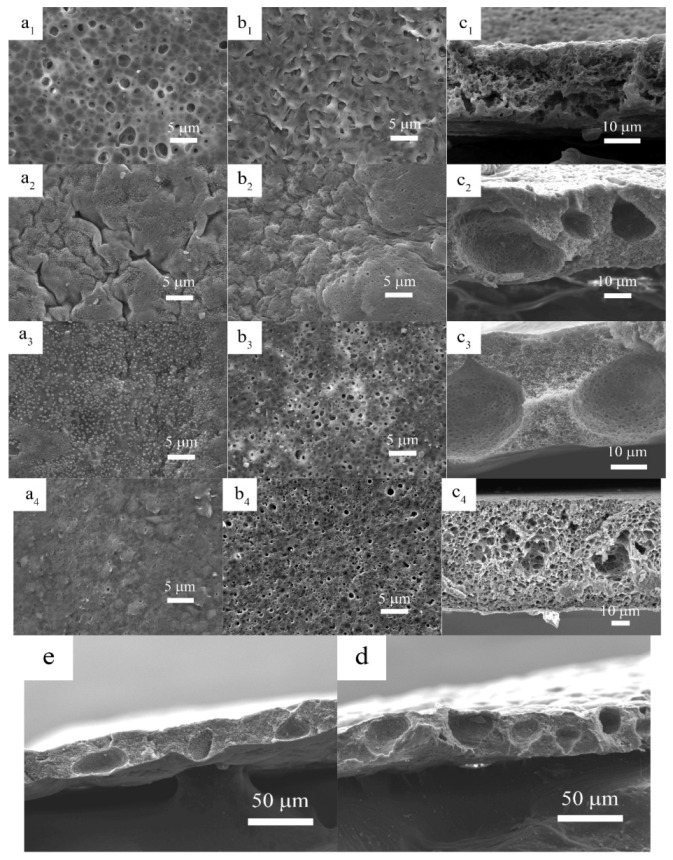
The surface and cross-sectional views of each group of membranes (**a**) shows the view of the upper surface; (**b**) that of the lower surface; and (**c**) shows the cross-sectional view; 1, 2, 3, and 4 are the views when the additional amounts of OVAPOSS were 2%, 5%, 7%, and 0, respectively; (**d**,**e**) are the cross-sectional views of the membranes with 5% and 7% OVAPOSS added at relatively small magnifications.

**Figure 3 materials-14-02701-f003:**
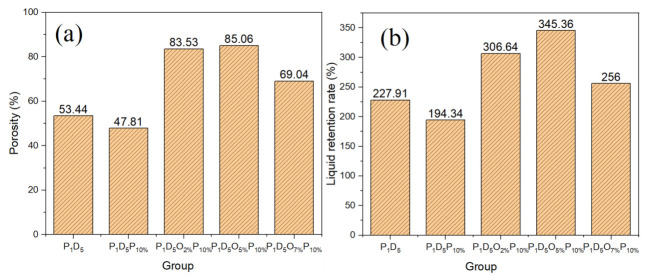
Porosity (**a**) and liquid absorption rate (**b**) of polymer membranes prepared with different OVAPOSS additions.

**Figure 4 materials-14-02701-f004:**
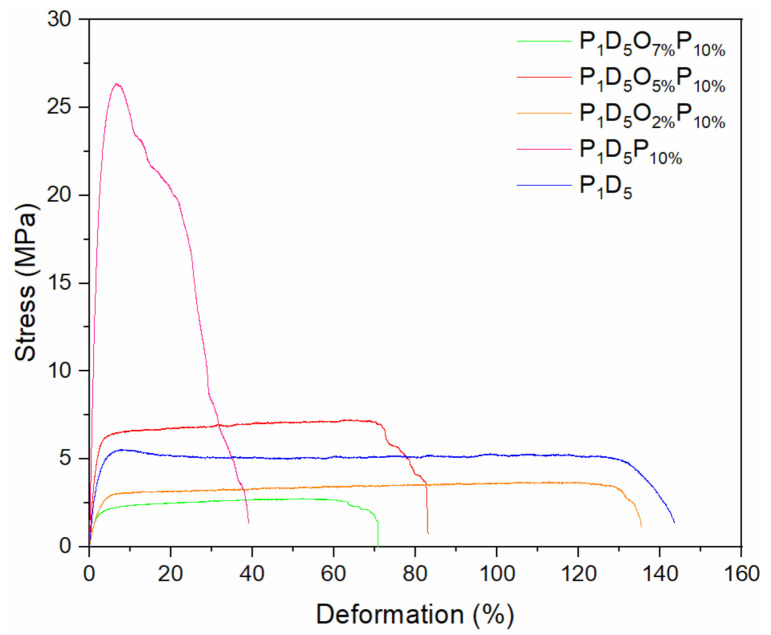
Strain–stress curves of polymer membranes prepared with different OVAPOSS additions.

**Figure 5 materials-14-02701-f005:**
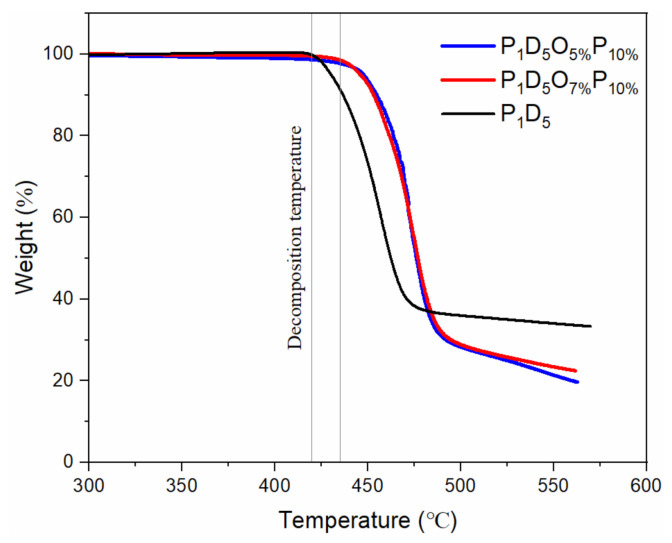
TG curve of the membranes.

**Figure 6 materials-14-02701-f006:**
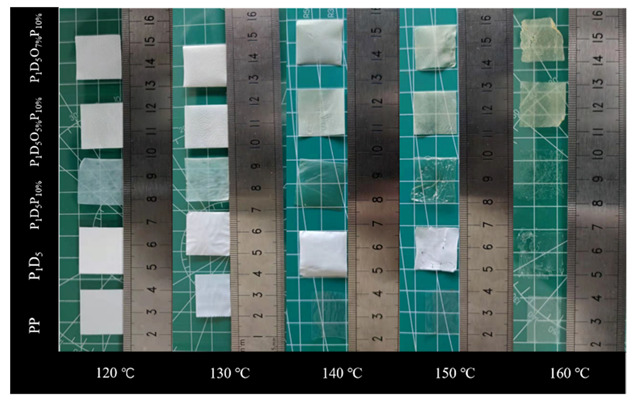
The thermal shrinkage pictures of each group of membranes at different temperatures.

**Figure 7 materials-14-02701-f007:**
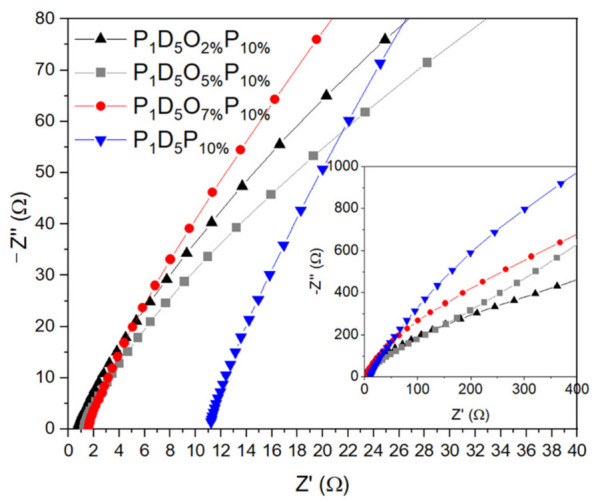
Room temperature impedance diagram of polymer membranes with different OVAPOSS doping amounts.

**Figure 8 materials-14-02701-f008:**
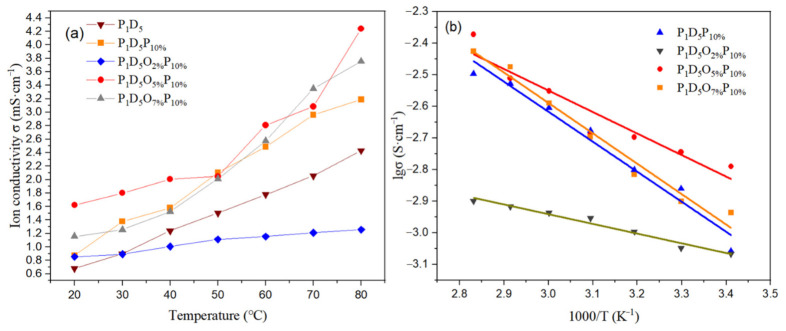
(**a**) Ionic conductivity diagram and (**b**) Arrhenius conductivity diagram of membranes prepared with different OVAPOSS contents at different temperatures.

**Figure 9 materials-14-02701-f009:**
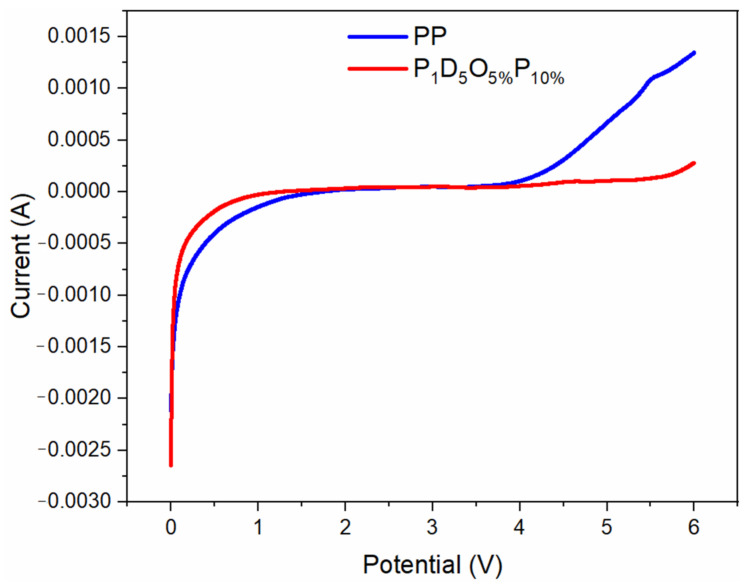
Linear sweep volt-ampere curve of membrane at room temperature.

**Figure 10 materials-14-02701-f010:**
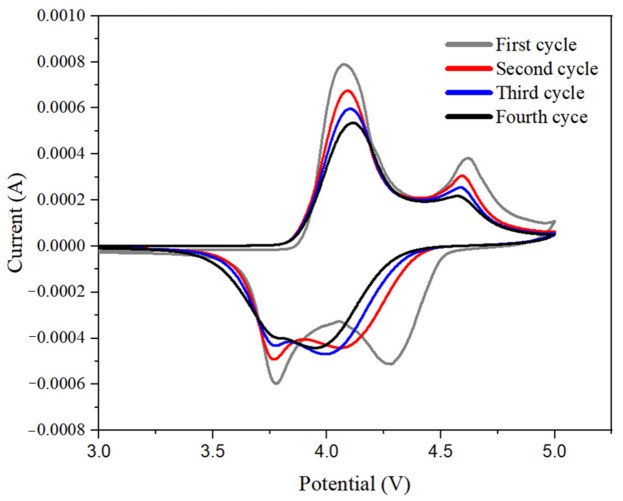
Cyclic voltammetry curve of Li_2_CoO_2_-Li half-cell assembled by P_1_D_5_O_5%_P_10%_ membrane.

**Figure 11 materials-14-02701-f011:**
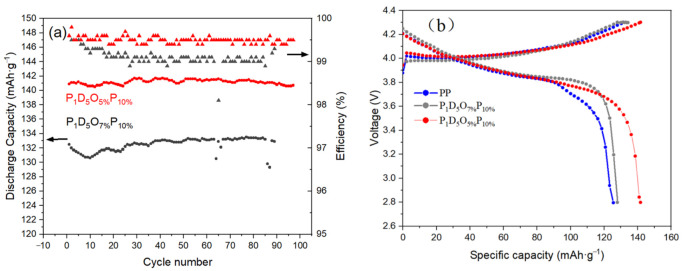
(**a**) Cycle performance; (**b**) Charge and Discharge curve of the first circle.

**Table 1 materials-14-02701-t001:** Room temperature impedance and conductivity of polymer membranes with different OVAPOSS doping amounts.

OVAPOSS Doping Amount	Thickness (μm)	Impedance (Ω)	Room Temperature Conductivity (×10^−3^ S·cm^−1^)
P_1_D_5_O_2%_P_10%_	14	0.835	0.855
P_1_D_5_O_5%_P_10%_	29	0.913	1.62
P_1_D_5_O_7%_P_10%_	34	1.5	1.16
P_1_D_5_P_10%_	189	11.03	0.874
P_1_D_5_	41	3.064	0.683

**Table 2 materials-14-02701-t002:** Different temperature ion conductivity of polymer membranes prepared with different OVAPOSS doping amounts.

Temperature (°C)		20	30	40	50	60	70	80
Ion conductivity (×10^−3^ S·cm^−1^)	P_1_D_5_P_10%_	0.874	1.378	1.580	2.105	2.487	2.962	3.189
	P_1_D_5_O_2%_P_10%_	0.855	0.893	1.007	1.113	1.156	1.211	1.258
	P_1_D_5_O_5%_P_10%_	1.621	1.800	2.01	2.046	2.808	3.082	4.239
	P_1_D_5_O_7%_P_10%_	1.156	1.255	1.527	2.01	2.574	3.349	3.755

**Table 3 materials-14-02701-t003:** Activation energy of membranes prepared with different OVAPOSS doping levels.

Group	P_1_D_5_	P_1_D_5_P_10%_	P_1_D_5_O_2%_P_10%_	P_1_D_5_O_5%_P_10%_	P_1_D_5_O_7%_P_10%_
Ea/kJ·mol^−1^	7.82	7.91	2.56	5.67	8.00

## Data Availability

Data is contained within the article.

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
