# Peer review of "Performance Improvement of PVDF–HFP-Based Gel Polymer Electrolyte with the Dopant of Octavinyl-Polyhedral Oligomeric Silsesquioxane"

_materials, 2021, doi:10.3390/ma14112701_

Round 1

Reviewer 1 Report

The manuscript “Performance Improvement of PVDF-HFP Based Gel Polymer Electrolyte with the Dopant of Octavinyl Cage Silsesquioxane” deals with the production of PVDF-HFP based gel polymer membrane, by phase inversion, at enhanced properties thanks to the addition of octavinyl cage silsesquioxane. This material was designed to be used as an electrolyte for lithium ion batteries; for this purpose, polyvinylpyrrolidone was also loaded in the polymeric matrix to improve the overall porosity.

The idea is ambitious and the results of the several analyses performed on these materials were intriguing. The work is in line with the aims of Materials.

Detailed comments:

- Introduction. The state of the art on the use of PVDF-HFP gels in this research field can be enlarged; for this purpose, see, for instance, the work of Sarno et al., SC-CO2-assisted process for a high energy density aerogel supercapacitor: The effect of GO loading, Nanotechnology, 2017, 28, Article number 204001.

- Results. A table reporting the pore mean size and the porosity values, depending on the concentration of OVAPOSS, can help the reader in understanding the effect of the additive on the membrane morphology.

- English can be improved.

Reviewer 2 Report

The manuscript, “Performance Improvement of PVDF-HFP Based Gel Polymer Electrolyte with the Dopant of Octavinyl Cage Silsesquioxane” reports a gel polymer electrolyte based on modified PVdF for application in Li-ion batteries. The gel polymer electrolyte appears to have a high ionic conductivity at room temperature and shows an excellent cyclability in a full cell format with LiCoO2 cathode. The results are noteworthy, and the material and electrochemical characterization reported here is sufficient. However, there are a few missing details and minor concerns that need to be addressed, after which this manuscript could be suitable for publication in Materials. They are as follows:

  1. In Table 2, the name of polymer membranes in third and fourth row are the same (P1D5 O7%P10%). Is this correct? I think one of them should be P1D5 O5%P10% instead. The authors need to correct this, if so.
  2. In Figure 7, in the Nyquist plot, could the authors also show the zoomed-in portion of the intercept? It is hard to observe and compare the resistance values with the scale that is currently shown in the manuscript.
  3. In Figure 11a, could the authors show the Coulombic efficiency data as well, for both the polymer membranes? This would shed light on their fading trends (if any).
  4. I would urge the authors to rewrite the Methods (Sections 2.2, 2.3, 2.5) section in passive voice format instead of the active voice format that the manuscript currently has.
  5. In Page 1, Abstract, Line 24, it should be “… at a rate of 1C” and not “ratio of 1C”. I would also urge the authors to thoroughly proof-read the manuscript for errors in English and grammar.

Reviewer 3 Report

The manuscript titled “performance improvement of PVDF-HFP based gel polymer electrolyte with the dopant of octavinyl cage silsesquioxane ’’ by Guo et al. reported that PVDF-HER based gel polymer electrolyte with PVP pores and octavinyl cages silsesquioxane for energy storage applications especially in Li-ion battery. This work is interesting and systematically studied polymer gel electrolyte membrane and used in Li-ion battery applications. Therefore, based on the merits, I am recommending this manuscript for publication in Materials. However, before the publication the following comments need to be thoroughly addressed.

 Minor revision:

  1. Author should carefully recheck the typo and grammatical errors in manuscript.
  2. Figure 2 captions are not clear and confusing. Therefor authors can revise the figure captions clearly.
  3. Why authors used for PVA in the gel electrolyte prepared and what is the important role?
  4. How to author confirmed doing of OVAPOSS with PVDF-HFP? Explain.
  5. The conclusion part should be more elaborate towards energy storage applications.
  6. Some of the important references are need to cite about the polymer electrolyte for energy storage applications in revised introduction part: 1002/app.49993 and 10.1007/s11581-013-0985-z

Round 2

Reviewer 1 Report

The authors improved the manuscript. The publication is recommended.

Reviewer 2 Report

The authors have addressed the comments and revised the manuscript accordingly. 

Reviewer 3 Report

Current form of the manuscript is now acceptable for publication in Materials.